# Green Extraction of Antioxidant Fractions from *Humulus lupulus* Varieties and Microparticle Production *via* Spray-Drying

**DOI:** 10.3390/foods12203881

**Published:** 2023-10-23

**Authors:** Tania Ferreira-Anta, María Dolores Torres, Jose Manuel Vilarino, Herminia Dominguez, Noelia Flórez-Fernández

**Affiliations:** 1CINBIO, Universidade de Vigo, Departamento de Ingeniería Química, Facultad de Ciencias, 32004 Ourense, Spain; tania.ferreira@uvigo.gal (T.F.-A.); matorres@uvigo.gal (M.D.T.); noelia.florez@uvigo.gal (N.F.-F.); 2R&D Department, Hijos de Rivera, S.A.U., 15008 A Coruña, Spain; jmlopez@estrellagalicia.es

**Keywords:** hop, mannitol, pressurized hot water, ultrasound-assisted extraction, viscosity

## Abstract

The formulation of polymeric microparticles to encapsulate bioactive compounds from two hop varieties (Nugget and Perle) using sequential green extraction processes was performed. The technologies used were ultrasound-assisted extraction (UAE) and pressurized hot water (PHW) extraction. Liquid phases were analyzed for total phenolic content (~2%), antioxidant activity (IC_50, DPPH_: 3.68 (Nugget); 4.46 (Perle) g/L, TEAC (~4–5%), FRAP (~2–3%), and reducing power (~4%)), protein content (~1%), oligosaccharide content (~45%), and for structural features. The fractions obtained from UAE were selected to continue with the drying process, achieving the maximum yield at 120 °C (Perle) and 130 °C (Nugget) (~77%). Based on these results, the formulation of polymeric microparticles using mannitol as the carrier was performed with these fractions. The production yield (~65%), particle size distribution (Perle: 250–750 µm and Nugget: ~100 µm), and rheological features (30–70 mPa s at 0.1 s^−1^) were the parameters evaluated. The UAE extracts from hop samples processed using a sustainable aqueous treatment allowed the formulation of microparticles with a suitable yield, and morphological and viscosity properties adequate for potential food and non-food applications.

## 1. Introduction

Biopolymer-based microparticles are useful to protect highly valuable compounds that can have therapeutic or food uses; moreover, the features of particulate systems allow to modulate drug delivery, and can improve the appearance, texture, taste or stability of food products [1,2]. Currently, the formulation of particulate systems using antioxidant compounds obtained from natural sources by sustainable processes has been gaining attention [3,4]. Microencapsulation via spray-drying is one of the most employed techniques; it favors fast solvent evaporation allowing the production of spheric particles [5]. In comparison to other conventional procedures, microencapsulation has the advantage of producing microparticles in a quite simple continuous operation [6]. In order to formulate these systems, food-grade biopolymers, such as mannitol or alginate, are used [2,7,8]. Their physicochemical features, like molecular weight, branching, solubility, charge or polarity, enable the assembly of colloidal particles, achieving a range of functional attributes.

*Humulus lupulus* (commonly known as hops), is a climbing plant belonging to the Cannabaceae family. This raw material could be an attractive alternative natural source to cope with increasing market demand for antioxidants [9,10]. *H. lupulus* has been used in traditional medicine due to its properties including sedative, antibacterial, antifungal, and antimicrobial activity [11,12]. Currently, the mainly use of hops is associated with the beer industry, the most relevant compounds being essential oils, volatile compounds, and bittering acids [13,14,15]. In addition, several studies have shown health benefits of the polyphenolic components, such as antioxidant, antiseptic or antitumoral properties [16,17]. Recently, the recovery of the phenolic compounds from hops has been studied with a focus on their incorporation as innovative ingredients for food, pharmaceutic or cosmetic applications [18].

To produce bioactive extracts from hop varieties, the effectiveness of conventional processes has been gradually enhanced with the assistance of intensified techniques, namely, the employment of greener solvents. The current industrial challenge is the design of extraction processes to produce several products according to the biorefinery target [14]. Green extraction technologies like pressurized hot water (PHW), microwave- or ultrasound-assisted extraction (UAE) have several advantages, including saving time, energy, and resources, allowing high efficiency and low environmental impacts [19,20]. Ultrasound-assisted extraction, also known as sonication, is used as a green extraction technology in order to obtain bioactive compounds from hops. This technology can promote the release of bioactive compounds, whereas the application of hydrothermal treatments leads to enhanced phenolic extraction yield [21]. Other authors evaluated the cascade combination of ultrasound- and microwave-assisted extraction to obtain phenolic compounds [22].

In this context, the major aim of this study is to explore the potential of the aqueous crude extracts obtained from two hop varieties (Nugget and Perle Hallertau) by a sequence of green extraction technologies (pressurized hot water and ultrasound-assisted extraction) using only water as the extractive agent. The bioactive fractions selected were the base to produce biopolymeric particulate systems using mannitol as the carrier. A protective layer of their highly valuable compounds with antioxidant properties was studied.

## 2. Materials and Methods

### 2.1. Raw Material

Two hop sample varieties, Nugget (N) and Perle (P), in pellet format, were kindly provided by Hijos de Rivera S.A. (A Coruña, Spain). The pellets were ground to a powder consistency and stored in darkness at room temperature (~25 °C) and RH ~40%, in hermetic plastic bags. The detailed composition was previously reported [22]: moisture 9.8% (P) and 8.7% (N), protein 16.1% (P) and 16.5% (N), ashes 6.8% (P) and 7.0% (N), and total carbohydrates 31.6% (P) and 32.6 (N).

### 2.2. Extraction Sequential Process

The sequential extraction process allows the recovery of the maximum quantity of bioactive compounds obtaining different fractions with different possible applications and an exhausted solid.

#### 2.2.1. Ultrasound-Assisted Extraction

The powdered hops were mixed with distilled water as solvent in a flask (solid:liquid ratio 1:15, *w*/*w*) and introduced in a water bath (FB 11207, Fisherbrand, Singen, Germany). The ultrasound-assisted extraction (UAE) was carried out using distilled water as solvent (solid:liquid ratio 1:15, *w*/*w*). The operation conditions were optimized in a previous work [22] using a different solvent to extract the bioactive compounds. The operational conditions were frequency at 80 kHz and amplitude at 100%, in sweep mode; the extraction time for Perle was 30 min and for Nugget it was 120 min; the temperature of the extraction was 55 ± 2 °C. After the extraction process, two phases (liquid and solid) were obtained via vacuum filtration, and both were characterized.

#### 2.2.2. Pressurized Hot Water Extraction

After the UAE treatment, the residual solids of both varieties of hops were introduced in a stainless steel pressurized and stirred reactor (Parr Instruments, series 4848, Moline, IL, USA). The extraction temperature (190 °C) was selected according to previous work where microwave-assisted extraction was used as the green extraction technology [22]. When the reactor achieved the selected temperature (190 °C), it was cooled quickly to room temperature in order to open it. Next, the liquid and solid phases were separated via vacuum filtration and characterized.

### 2.3. Solid Residue Fraction Characterization

The raw material and solid residues separated by vacuum pump were characterized after the sequential extraction processes, at least in triplicate.

The moisture content in the solid sample was determined according to the standard gravimetric method developed by the Association of Official Analytical Chemists (2000). The sample was maintained for 48 h in an convective air oven at 105 ± 2 °C (P Theroven, JP Selecta, Abrera, Spain). The ash content was calculated per 100 g of raw material, and this content was determined gravimetrically using a muffle furnace (ELF, Carbolite, Hope Valley, UK) at 575 °C for 6 h. The protein content present in the two hop varieties was determined by the Kjeldahl method and calculated applying the conversion factor (6.25) on the total nitrogen content determined according to the AOAC protocol (2000).

#### 2.3.1. Minerals

The minerals of the solid fractions were analyzed and determined as macroelements (Na, K, Ca, Mg, Fe) and microelements (Cd, Pb, Cu, Hg) [23]. In the case of mercury, the content was evaluated using cold vapor atomic absorption spectrometry following the protocols described elsewhere [23].

#### 2.3.2. Oligosaccharide Fraction

The oligosaccharide fraction was performed using sulfuric acid at 72% (acid hydrolysis). This procedure was performed in a water bath at 30 °C for 1 h with manual stirring. Immediately, a second step with sulfuric acid at 4% was carried out, at 121 °C for 1 h at 2 atm. The liquid phase was filtered and analyzed using high-performance liquid chromatography (HPLC) (1260 Infinity, Agilent Technologies, Santa Clara, CA, USA), following the protocol described elsewhere [24]. In the case of the solid phase, it was evaluated gravimetrically as acid insoluble residue (AIR).

### 2.4. Liquid Fraction Characterization

The dried content of the soluble extract was determined after drying 1 mL at a constant temperature of 105 °C in an air convective oven (P Theroven, JP Selecta, Spain), in triplicate, for 72 h. The pH value was experimentally measured using a pH meter (Crison GLP 21, Crison, Alella, Spain) previously calibrated using standards (Crison, Alella, Spain).

The soluble protein content was determined using the Bradford method [25]. The protocol to measure the quantity of protein in the samples was performed following the instructions provided by the supplier. Bovine serum albumin (BSA, Sigma, Castellón de la Plana, Spain) was the standard used to perform the standard curve; the absorbance was measured at 595 nm. The protocol was carried out at least in triplicate.

The oligosaccharide fraction was determined following the protocol described above [24]. Previously, a hydrolysis of the liquid samples was performed (4% sulfuric acid, 121 °C, 20 min).

#### 2.4.1. Antioxidant Features

The total phenolic content (TPC) in each extract was determined using the Folin–Ciocalteu spectrophotometric method [26]. The Folin–Ciocalteu reagent (VWR Chemicals, France) was diluted 1:1 (*v*/*v*) with distilled water and sodium carbonate solution at 10% (Na_2_CO_3_, Sigma-Aldrich, Burlington, MA, USA). The standard curve was prepared using gallic acid (Sigma Aldrich, Shangai, China) in concentrations from 0.01 to 0.1 ppm. For the measurements, 0.25 mL of sample, 1.875 mL of distilled water, 0.125 mL of Folin reagent (previously prepared), and 0.25 mL of sodium carbonate 10% were introduced in test tubes, homogenized in a vortex, and incubated for 1 h in darkness at room temperature. The absorbance was read at 765 nm using a spectrophotometer (Evolution 201, Thermo Scientific, Bremen, Germany). The determinations were carried out in triplicate.

#### 2.4.2. Antioxidant Activity

The antioxidant capacity was evaluated using the Trolox equivalent antioxidant capacity (TEAC) assay [27]. Briefly, the liquid samples and water for the control (10 μL) were placed in a test tube. The ABTS solution was diluted with PBS until absorbance 0.7 ± 0.1 at 734 nm, and was added to 1 mL. The test tubes were incubated for 6 min at 30 °C in a water bath, and the absorbance was read at 734 nm. The standard curve was prepared using Trolox (Sigma-Aldrich, Søborg, Denmark), and the data were expressed as TEAC values.

The ferric reducing antioxidant power, also known as FRAP, is based on the reduction of the ferric 2,4,6-tripyridyl-s-triazine (TPTZ) complex in acidic conditions [28]. The FRAP solution was prepared with 10 mM TPTZ and 40 mM HCl. The sample or standard was placed in a test tube (0.1 mL), and 3.0 mL of FRAP solution was added. The absorbance was read at 593 nm for 6 min, and the assay was performed at least in triplicate. Ascorbic acid was used as standard.

The reducing power is determined as the reduction of Fe (III) into Fe (II). The hop extracts were placed in a test tube (1 mL), and phosphate buffer at 0.2 M, pH 6.6 (2.5 mL), was added. Afterward, potassium ferricyanide at 1% (*w*/*v*) (2.5 mL) was added. The test tube was mixed in a vortex and incubated (50 °C, 30 min). Trichloroacetic acid at 10% (2.5 mL) was added, and the mixture was centrifuged. The supernatant obtained was mixed with distilled water at a ratio 1:1 (*v*/*v*), and ferric chloride at 0.1% (0.5 mL) was added. The absorbance was measured at 700 nm, at least in triplicate. The standard curve was prepared used ascorbic acid as standard.

The radical scavenging activity or DPPH (α,α-diphenyl-β-picrylhydrazyl) of the samples was evaluated [29]. The hop extracts were placed in test tubes (50 μL) and DPPH solution, previously prepared and adjusted to 0.6 absorbance (measured at 515 nm), was added. The percentage of inhibition (IC_50_) was determined and expressed as g/L.

All analyses mentioned above were performed at least in triplicate.

#### 2.4.3. Structural Profiles

##### High-Performance Size-Exclusion Chromatography

The molar mass distribution of saccharide samples was analyzed by high-performance size-exclusion chromatography, using the HPLC described above. The columns used were TSKGel G2500PW_XL_ and G3000PW_XL_, in series (300 × 7.8 mm, Tosoh Bioscience, Griesheim, Germany), and a PWX-guard column (40 × 6 mm, Tosoh Bioscience, Griesheim, Germany). The mobile phase was Milli-Q water, at 0.4 mL/min. The standards used were dextrans from 1000 to 80,000 g/mol (Fluka, St. Louis, MO, USA).

##### Fourier-Transform Infrared Spectroscopy

Hop lyophilized samples from the Perle and Nugget varieties and the microparticles formulated were blended with KBr and dried with an infrared lamp for 30 min. FTIR spectra were recorded at 400–4000 cm^−1^ at 25 scans/min (Bruker IFS 28 Equinox equipment, OPUS-2.52, Billerica, MA, USA) for data acquisition using System 450-MT2.

### 2.5. Production of Microparticles

The equipment employed to produce microparticles from both studied varieties was a mini spray-dryer B-290 (BÜCHI, Flawil, Switzerland) equipped with a standard and high-yield cyclone, equipped with a 1.5 mm nozzle. The tested parameter was the inlet temperature (from 110 to 150 °C), and the flow rate was fixed at 0.7 mL/min, with 20% feed solution. The atomization air flow rate was evaluated from 439 to 1052 L/h.

The polymer used as the carrier was mannitol, widely used for pharmaceutical applications due to its high compatibility and inert properties. In order to establish the operation conditions to formulate the microparticles, the UAE extract from both varieties was dried using mannitol (1, 2, 5 and 10%, *w*/*w*) as the carrier at 120 °C and 130 °C for Perle and Nugget, respectively.

#### 2.5.1. Characterization of Microparticles

##### Yield of Production

The microparticle production yield (%) was calculated gravimetrically by weighing the microparticles found in the collector, following Equation (1):(1)$$Production\;yield\left(\%\right)=\frac{mg\;microparticles\;recovered}{mg\;extract+mg\;mannitol}\cdot 100$$

##### Scanning Electron Microscopy

The morphology of the microparticles obtained via atomization was analyzed using scanning electron microscopy (SEM, JEOL JSM6010LA, Tokyo, Japan). Previously, the microparticles were covered with a gold layer of 15 nm, and several images were acquired at different scales to evaluate several parameters.

##### Particle Size Distribution

This parameter of the formulated microparticle samples was analyzed from the images obtained via SEM using the ImageJ software (V 1.8.0.). The measurements of microparticle diameters were made manually, and up to 100 measures per microparticle formulation were evaluated.

### 2.6. Rheological Measurements

Steady-state shear tests of selected microparticulate systems were performed at 25 °C on a MCR 302 controlled-stress rheometer (Paar Physica, Graz, Austria). The measuring system consisted in a sand-blasted parallel plate (25 mm diameter, 0.5 mm gap). The apparent viscosity versus the shear rate of selected microparticulate dispersions (2%) typically used for drug delivery systems was monitored in the range of shear rates from 0.1 to 100 s^−1^. The above content is in the range of that typically employed for natural materials in different fields from the food to the cosmetic industry [30]. The time-dependent shear thinning was also monitored by measuring the forward and backward apparent viscosity curves. In all cases, light silicon oil was employed to cover the edge of the dispersions to avoid water evaporation during measurements. It should be noted that the samples were allowed to rest for 10 min in the measuring geometry before rheological testing to promote the structural and thermal equilibration of the dispersions.

### 2.7. Statistical Analysis

In all cases, measurements were performed at least in triplicate. Statistical study was carried out by means of one-factor analysis of variance. The Scheffé test was conducted to differentiate means values if the ANOVA showed means differences. For this purpose, a degree of confidence of 95% (*p* < 0.05) was employed using the PASW Statistics v.22 software (IBM SPSS Statistics, New York, NY, USA).

## 3. Results and Discussion

### 3.1. Sequential Extraction Process

This work was focused on sequential extraction using two varieties of hops as raw material, known as Perle and Nugget. The main aim of this strategy was to obtain the maximum quantity of bioactive compounds. An overview of the processing and the main characterization of both materials is presented in Figure 1. The extraction technologies selected were ultrasound-assisted extraction (UAE) and pressurized hot water (PHW) extraction. The combination of both was labeled as UAE-PHW.

The sequential extraction process from Perle and Nugget has a notable influence on the results of characterization; i.e., the total carbohydrate content showed an increase after the extraction treatment, suggesting that this process continues until the complete extraction of carbohydrates. Moreover, the severity of the PHW process could influence the mineral content decreasing after the extraction process. Several works have found differences in bioactive and biological properties of the extracts using different extraction technologies [10,14,18]. This work was based on the biorefinery concept with the aim to reduce waste. The solid residues obtained in the first step of the extraction process were used in the next step to achieve the maximum quantity of bioactive compounds from the raw material, the *H. lupulus* varieties Perle and Nugget. In addition, the solvent and operational conditions used in the extraction will have an influence on the results [14,22,31].

### 3.2. Characterization of Solid Residue Fraction

Table 1 shows the results of the proximal composition of the solid phases after the extraction process. The major quantity of ashes, carbohydrates, and minerals were extracted in the first step (UAE), and the sequential extraction treatment (UAE-PHW) also has an influence on the results. The ash content decreased in both varieties whereas the acid insoluble residue (AIR) increased. Minimum differences in total carbohydrates were found in the case of the Nugget variety. The other variety, Perle, showed around 4% of rhamnose after the second treatment (UAE-PHW). The content of protein in all residual fractions of hops achieved values around 13–14%, whereas the content of total carbohydrates for Nugget and Perle was 7% and 5%, respectively.

The varieties and origin of samples could be the reason for the different results obtained. Other authors found that the composition of spent hops and brewing residues was consistent with the present results, except for the higher protein content (20–70%) [10]. Both hops showed a value for the mineral content of around 1.6–1.7% after the UAE process, decreasing to 1.3% after the UAE-PHW extraction treatment (Table 2). A previous work has reported similar results [10]. The main macroelements were potassium, calcium, and magnesium with phosphorus in a minor proportion [10]. The solvent, the extraction process selected, and the severity could be causing these differences [14].

#### 3.2.1. Liquid Fraction Characterization

##### Antioxidant Activity

Figure 2a shows a similar trend for both varieties, with a slight increase after the two extraction technologies; the values increased except the total phenolic content and the reducing power in Perle. The value of TEAC after the sequential extraction process increased from 5.2 to 6.1 for Perle, and from 2.6 to 4.2 for the Nugget variety. The higher content for the Perle variety could suggest higher solubility of this raw material in comparison with Nugget. The antioxidant activity was also evaluated using a DPPH assay, and the minimum value for IC_50_ (g/L) was identified for UAE-PHW Nugget (3.68), followed by Perle (UAE = 4.46; UAE-PHW = 4.76) and UAE Nugget (5.73), suggesting lower activity than those of butylhydroxyanisol and butylhydroxytoluene.

Other authors assessed the total phenolic content in extracts obtained from hops and reported a higher value using ethanol in comparison with water; similar results were also found for TEAC value [32]. These results suggest that organic solvents promote the recovery of phenolic content from this raw material. In the current work, the proposed extraction promoted the recovery of the antioxidant fraction from the Nugget variety, but no significant differences were found for the Perle variety. Sanz and coauthors (2022) evaluated the features of the ethanolic fraction obtained from *H. lupulus*, observing a similar performance with the current work regarding DPPH and ABTS antiradical capacities [22,32].

The total carbohydrate content is presented in Figure 2b. Values around 30% for the liquid fraction obtained after UAE and around 45% for UAE-PHW extraction were found. As expected, this content was higher than that obtained in the extraction with organic solvent (ethanol) with values between 3 and 10% [22]. This behavior suggests that water extraction could be a suitable strategy to obtain saccharide fractions when *H. lupulus* is used as the raw material. On the other hand, the glucose fraction obtained using UAE was around 13.5% for both varieties. In the case of the samples obtained after the sequential extraction process, the maximum was observed for xylose, with a value 13.5% for both varieties, confirming the solubilization of the hemicellulosic fraction.

### 3.3. Structural Profiles

#### Molecular Weight Profile

The molar mass distribution profile of the liquid samples obtained using the tested extraction process were evaluated. Figure 3 displays these results. When UAE extraction was performed, two fractions were differentiated (Figure 3a). In the case of Nugget, one fraction was between 1000 and 5000 Da, and the other under 1000 Da. A similar behavior was observed with the Perle variety for the small size fraction; the other peak was found around 5000 Da.

After the sequential extraction (UAE-PHW), significant differences between the varieties were observed (Figure 3b). The profile of Perle showed three fractions, under 1000 Da, between 25,000 and 5000 Da, and above 80,000 Da, whereas the Nugget variety only displayed a fraction under 1000 Da. According to these results, the extraction process had an influence on the molar mass distribution profile, since the higher severity of the pressurized hot water extraction causes hydrolytic actions on the cell walls.

These results were in line with another work where fractions with different molecular weights were obtained [33]. The extraction technology applied has an important role in the molecular mass distribution profile of the samples; the severity and other factors suggest the influence in these results [34].

### 3.4. Fourier-Transform Infrared Spectroscopy

The bands of the representative groups present in *H. lupulus* and mannitol are shown in Figure 3. The samples analyzed were the extracts dried via atomization with the highest yield (120 and 130 °C for Perle and Nugget, respectively), and the representative microparticles (MP) formulated with mannitol as the polymeric carrier, at different percentages.

The extracts from Perle (Figure 3c) and Nugget (Figure 3d) were analyzed, and the peaks found between 985 cm^−1^ and 1025 cm^−1^ could be vibrations of the CH_2_-OH groups of carbohydrates [35]. The bands found at 1373 and 1400 cm^−1^ suggest stretching vibrations of C–H and CH_2_ groups; usually they are observed between 1300 and 1460 cm^−1^. The peaks found at 1608 cm^−1^ could be related to phenolic ring C=C, detected in the range between 1508 and 1653 cm^−1^ [36,37].

Mannitol and the corresponding microparticles (Figure 3) were also analyzed to explore the main peaks. In this case, bands at 1053 and 1070 cm^−1^ were found in the spectrum, and these peaks could be associated with the C-O stretching group [38]. The peak obtained at 875 cm^−1^ suggests being highly dependent on the presence of mannitol.

Some studies have reported the utilization of different hop varieties, evaluated to extract bioactive compounds [36,37].

#### Microparticle Production and Characterization

The liquid samples obtained after UAE treatment were dried via atomization, and temperatures from 110 to 150 °C were tested (Figure 1). In the case of Perle, the yield ranged from 64 to 77% (*w*/*w*), and for Nugget from 66 to 77% (*w*/*w*), these being the maximum values obtained at 120 and 130 °C, respectively. Table 3 shows the yields. Based on these results, polymeric microparticles using different percentages of mannitol (0–10%, *w*/*w*) were produced. The content of mannitol influences the drying yield of the microparticles. In the case of the Perle variety, the values have shown a maximum yield of 66.8% using 1% of mannitol, whereas the maximum value for Nugget was 63.7 at 2% of mannitol. No significant differences at 2% mannitol for the two varieties were observed. 

The behavior observed suggests that the mannitol percentage is an important factor. Nevertheless, the values obtained using 1% (*w*/*w*) were suitable and similar to other works [39]. When the percentage of mannitol was increased, the yield of microparticle production decreased. Therefore, the selection of the percentage of mannitol is a critical parameter depending on the final application. According to the antioxidant activity results and considering the production yield of the microparticles, the Perle variety features could be preferred in comparison to the Nugget variety.

### 3.5. Microstructure

The liquid samples obtained after the ultrasound-assisted extraction and sequential extraction process were dried via atomization, and the SEM of the microparticles is shown in Figure 4.

The morphological assessment of the formulated microparticles was performed using scanning electron microscopy. In both cases, the Perle and Nugget samples showed a similar appearance, and possible aggregates can be observed at a different magnification. The field vision of the samples was represented in the images to evaluate the behavior and features of the polymeric microparticle systems. Additional images of the polymeric microparticles produced at different mannitol contents for both studied hop varieties are presented in Appendix A.

The size distributions of the microparticles were also evaluated, observing the same behavior for both varieties. The extraction process influences the size; after the second extraction, the size of the microparticles increased noticeably.

The size of the microparticles obtained after UAE was 220–260 µm and 70–80 µm for Perle and Nugget, respectively. After the second extraction process (UAE-PHW), the size of the microparticles was 750 µm and 100–120 µm for Perle and Nugget, respectively. Other biopolymeric carriers could be evaluated to formulate several particulate systems and their features. In a work where alginate, starch, and carrageenan were evaluated, microparticles with different behaviors and shapes were obtained depending on the carrier used [8].

Analyzing the results achieved, the final application of the microparticles could be the reason to select the Nugget (bigger size) or Perle (smaller size) variety, since the values of antioxidant activity and phenolic compounds are greater for Nugget.

### 3.6. Rheology Measurements

Figure 5 shows the viscoelastic profiles of representative microparticulate systems measured at 25 °C, selected based on their spray-drying yields. The knowledge of their rheological properties is essential during processing to extend their possible applications. All dispersions exhibited non-Newtonian behavior, being the shear-thinning profiles successfully described by the well-known power law model (R^2^ > 0.989). The drop in the apparent viscosity over the tested shear rate range was similar for all tested dispersions (about 2 decades). At a fixed shear rate, the microparticles from Perle featured lower apparent viscosity when compared to those formulated with Nugget, independently of the selected extraction treatment. For both hop varieties, it was also observed that microparticles subjected to the sequential UAE-PHW processing presented lower viscosity than those made from ultrasound extracts, confirming the expected and previously mentioned hydrolysis of polysaccharidic fractions.

The rheological results suggested that microparticles incorporated with hop extracts with higher antioxidant potential feature smaller flow resistance. Nevertheless, the magnitude of the tested systems was consistent with those previously reported in the literature for polymer microparticles prepared with different natural sources [30,40]. It should be highlighted that the thixotropic character, with the consequent advantage for its application, was not identified in the tested systems at the studied experimental conditions.

## 4. Conclusions

Sequential extraction was performed to achieve the maximum recovery of bioactive compounds (phenolic compounds, antioxidant fraction, protein content, and carbohydrate content) in the liquid fraction from *H. lupulus* (varieties Perle and Nugget), with the highest content found for the Perle variety. The fractions obtained were suitable to produce polymeric microparticles using mannitol as the carrier. The formulations showed differences in the size distribution, with the highest size found for the Perle variety (750 µm) and the smallest for Nugget (120–150 µm). Hence, this is a critical parameter for future applications, because a higher content of bioactive compounds was exhibited for Perle. Moreover, the flow resistance of microparticles enriched with hop extracts notably decreased. Several fields of application could be suitable for these polymeric microparticles, from biomedical to cosmetic or nutraceutical, improving the stability of the final product or helping to release active compounds. Future studies should be focused on assays of toxicity, stability, and bioactive compound release to corroborate the potential of the microparticles during application, including the testing of other biopolymer carriers that could enhance their behavior.

## Figures and Tables

**Figure 1 foods-12-03881-f001:**
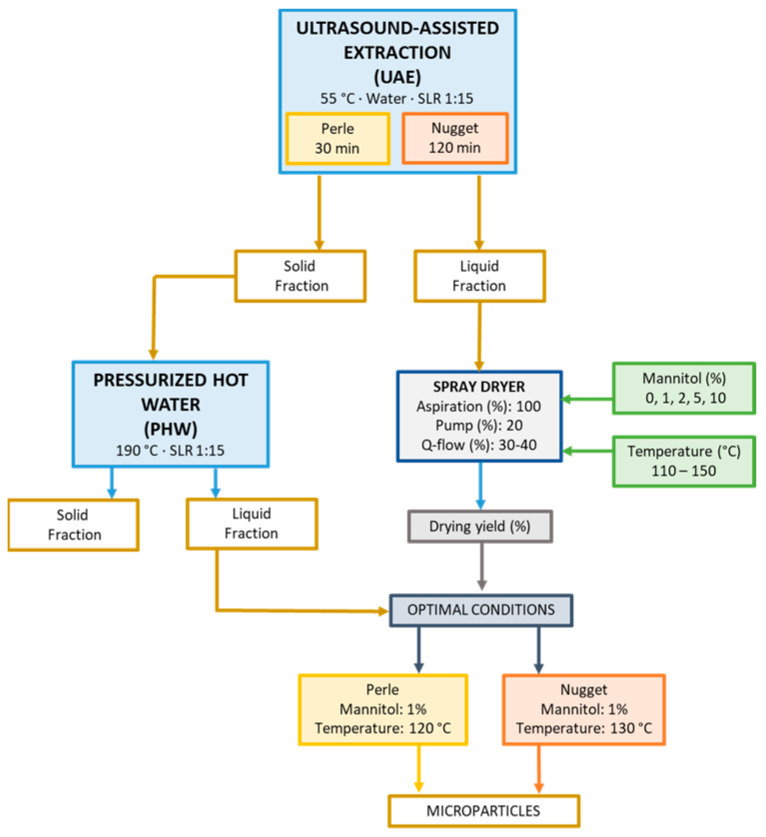
Flow diagram of sequential extraction using two varieties of *H. lupulus*, Nugget and Perle, as the raw material. Note: SLR means solid liquid ratio.

**Figure 2 foods-12-03881-f002:**
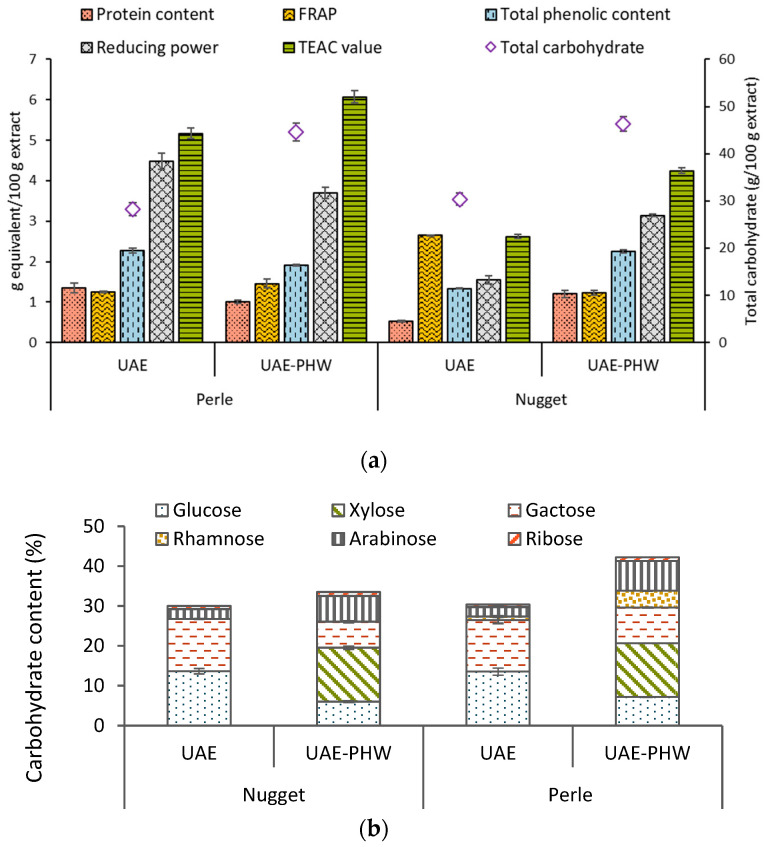
Total phenolic content, protein content, and TEAC value results (**a**) and carbohydrate content (**b**) for both varieties (Perle and Nugget), using ultrasound-assisted extraction (UAE) and the sequence of ultrasound-assisted extraction and pressurized hot water extraction (UAE-PHW). Note: TEAC: Trolox equivalent antioxidant capacity; FRAP: Ferric reducing antioxidant power.

**Figure 3 foods-12-03881-f003:**
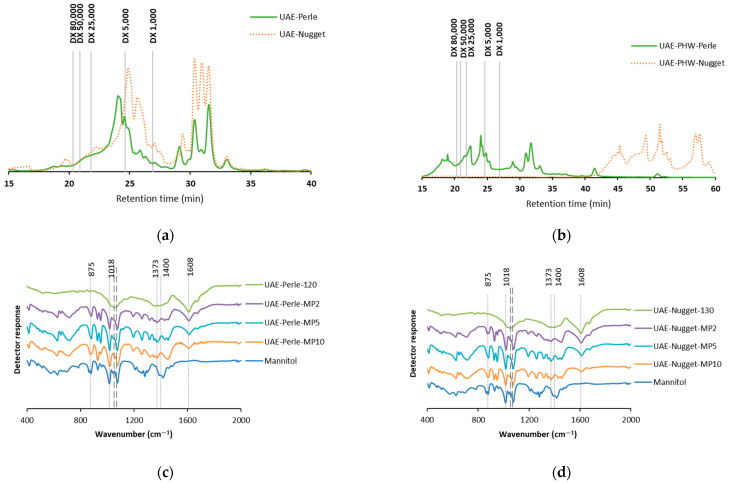
Profile of molar mass distribution of the liquid phases obtained after (**a**) ultrasound-assisted extraction (UAE) and (**b**) the sequential extraction UAE and pressurized hot water extraction (UAE-PHW), and spectra of FTIR of the extracts obtained after ultrasound-assisted extraction (UAE) and the microparticles (MP) formulated with mannitol at different contents: MP2 (2%), MP5 (5%), and MP10 (10%) for Perle (**c**) and Nugget (**d**). The solid line represents mannitol; the corresponding wavenumbers (1053 and 1070 cm^−1^) are plotted as dashed lines.

**Figure 4 foods-12-03881-f004:**
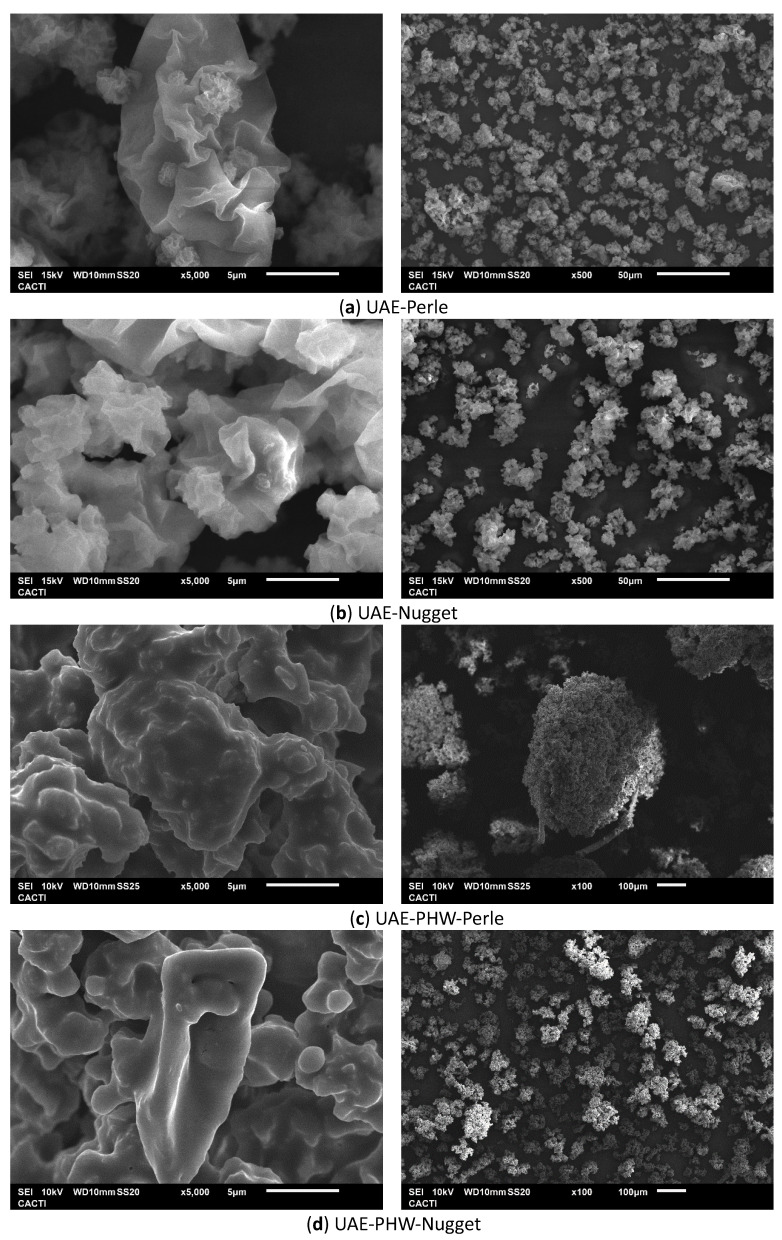
Scanning electron microscopy images of selected microparticles (1% mannitol) obtained after ultrasound-assisted extraction (UAE) and the sequential extraction process (UAEPHW) for the Perle and Nugget varieties from *H. lupulus*. Note: Magnitudes of the images of UAE Perle and Nugget: ×5000 and ×500; and images of UAE-PHW Perle and Nugget: ×5000 and ×100.

**Figure 5 foods-12-03881-f005:**
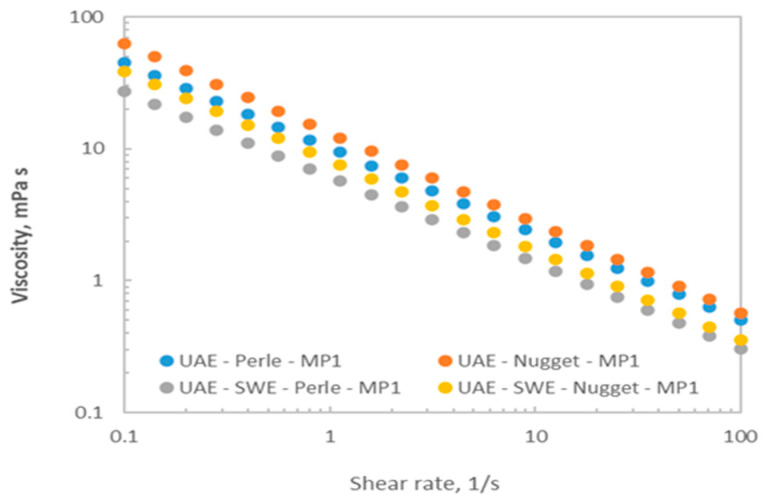
Steady shear measurements for microparticulate systems formulated at selected conditions with hop antioxidants coated with mannitol (1%).

**Table 1 foods-12-03881-t001:** Proximal composition of the solid fraction after ultrasound-assisted extraction (UAE) and after sequential ultrasound-assisted and pressurized hot water extraction (UAE-PHW) using two varieties of *H. lupulus*, Perle and Nugget, as the raw material.

	Perle		Nugget	
	UAE	UAE-PHW	UAE	UAE-PHW
*Moisture* (%)	6.47 ± 0.10 ^b^	5.22 ± 0.02 ^d^	7.92 ± 0.10 ^a^	5.64 ± 0.03 ^c^
*Ash* (%)	5.11 ± 0.21 ^a^	3.88 ± 0.14 ^c^	4.63 ± 0.14 ^b^	3.81 ± 0.09 ^c^
*Protein* (%)	13.53 ± 0.25 ^b^	13.57 ± 0.07 ^b^	14.21 ± 0.11 ^a^	14.11 ± 0.04 ^a^
*AIR* (%)	49.90 ± 3.86 ^b^	75.55 ± 1.00 ^a^	38.43 ± 2.16 ^c^	71.36 ± 3.86 ^a^
*Carbohydrates* (%)	5.39 ± 0.12 ^b^	4.79 ± 0.09 ^c^	7.05 ± 0.21 ^a^	7.08 ± 0.11 ^a^
Glucose	2.92 ± 0.04 ^c^	3.45 ± 0.04 ^b^	3.37 ± 0.06 ^b^	4.39 ± 0.03 ^a^
Xylose	1.23 ± 0.02 ^b^	0.96 ± 0.02 ^c^	1.60 ± 0.04 ^a^	1.26 ± 0.01 ^b^
Galactose	0.43 ± 0.02 ^a^	0.21 ± 0.02 ^c^	0.44 ± 0.02 ^a^	0.35 ± 0.01 ^b^
Rhamnose	0.16 ± 0.01 ^b^	3.94 ± 0.76 ^a^	0.24 ± 0.01 ^b^	0.09 ± 0.01 ^c^
Arabinose	0.40 ± 0.01 ^a^	0.17 ± 0.02 ^b^	0.41 ± 0.03 ^a^	0.20 ± 0.02 ^b^
Acetic acid	0.07 ± 0.01 ^d^	0.23 ± 0.01 ^c^	0.48 ± 0.01 ^b^	1.18 ± 0.59 ^a^

Superscripts with different letters by row show significant differences (*p* < 0.05).

**Table 2 foods-12-03881-t002:** Content of minerals in the solid fraction after ultrasound-assisted extraction (UAE) and after ultrasound-assisted and pressurized hot water extraction (UAE-PHW) from the two varieties.

	Perle		Nugget	
	UAE	UAE-PHW	UAE	UAE-PHW
*Total minerals* (%)	1.61	1.36	1.70	1.34
	Macroelements (g/kg)			
K	5.4	1.7	7.0	2.2
Ca	8.1	9.8	7.6	9.3
Mg	2.1	1.7	2.1	1.5
	Microelements (mg/kg)			
B	25	10.8	29	13.2
Cd	<2	<2	<2	<2
Cu	82.2	116	23.6	35
Na	310.4	325.6	311.8	310
Pb	<6	<6	<6	<6
Hg	<0.040	0.044	0.064	0.114

Macro- and microelements were analyzed in duplicate; the standard deviation was <10%. Data are given as mean ± standard deviations.

**Table 3 foods-12-03881-t003:** Yields of microparticle production using UAE extracts and mannitol as the carrier.

	Spray-Drying Yield (%)	
	UAE-Perle	UAE-Nugget
Mannitol/T (°C)	120 °C	130 °C
1	66.8 ^a^ ± 1.5	61.2 ^b^ ± 1.7
2	61.9 ^a^ ± 2.4	63.7 ^a^ ± 2.9
5	44.6 ^b^ ± 1.3	60.9 ^a^ ± 0.2
10	32.3 ^b^ ± 3.1	41.2 ^a^ ± 3.1

Data are given as mean ± standard deviations. Superscripts with different letters by row show significant differences (*p* < 0.05).

## Data Availability

The data presented in this study are available on request from the corresponding author.

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
