# Peer review of "Green Extraction of Antioxidant Fractions from Humulus lupulus Varieties and Microparticle Production via Spray-Drying"

_foods, 2023, doi:10.3390/foods12203881_

Round 1

Reviewer 1 Report

-The proximate, crabohydrate and mineral results in Table 1 should be shown in separate tables.

-Why was hop in pellet form turned into granules by spray drying?

-Why was the effect of hops, which is widely used in brewing, on the humulene and lupulene contents not investigated? Maybe the granule structure of spray drying may be good, but once the components that give flavor to the beer disappear, the properties of the granule structure are meaningless.

partly  should be revised

Author Response

Reviewer 1

-The proximate, carbohydrate and mineral results in Table 1 should be shown in separate tables.

Thank you for your comment, following your advice the authors have modified the table 1. Currently, Table 1 exhibits only information on the proximal composition and Table 2 shows the results on minerals.

Table 1. Proximal composition of the solid fraction after extraction processes ultrasound-assisted extraction (UAE) and sequential extraction process of ultrasound-assisted and pressurized hot water extraction (UAE-PHW) using as raw material two varieties of H. lupulus, Perle and Nugget.

Perle

Nugget

UAE

UAE-PHW

UAE

UAE-PHW

Moisture (%)

6.47 ± 0.10b

5.22 ± 0.02d

7.92 ± 0.10a

5.64 ± 0.03c

Ash (%)

5.11 ± 0.21a

3.88 ± 0.14c

4.63 ± 0.14b

3.81 ± 0.09c

Protein (%)

13.53 ± 0.25b

13.57 ± 0.07b

14.21 ± 0.11a

14.11 ± 0.04a

AIR (%)

49.90 ± 3.86b

75.55 ± 1.00a

38.43 ± 2.16c

71.36 ± 3.86a

Carbohydrates (%)

5.39 ± 0.12b

4.79 ± 0.09c

7.05 ± 0.21a

7.08 ± 0.11a

Glucose

2.92 ± 0.04c

3.45 ± 0.04b

3.37 ± 0.06b

4.39 ± 0.03a

Xylose

1.23 ± 0.02b

0.96 ± 0.02c

1.60 ± 0.04a

1.26 ± 0.01b

Galactose

0.43 ± 0.02a

0.21 ± 0.02c

0.44 ± 0.02a

0.35 ± 0.01b

Rhamnose

0.16 ± 0.01b

3.94 ± 0.76a

0.24 ± 0.01b

0.09 ± 0.01c

Arabinose

0.40 ± 0.01a

0.17 ± 0.02b

0.41 ± 0.03a

0.20 ± 0.02b

Acetic acid

0.07 ± 0.01d

0.23 ± 0.01c

0.48 ± 0.01b

1.18 ± 0.59a

Superscripts with different letter by row shows significant differences (p < 0.05).

Table 2. Content of minerals in the solid fraction after extraction processes ultrasound-assisted extraction (UAE) and sequential extraction process of ultrasound-assisted and pressurized hot water extraction (UAE-PHW) using as raw material two varieties of H. Lupulus, Perle and Nugget.

Perle

Nugget

UAE

UAE-PHW

UAE

UAE-PHW

Total minerals (%)

1.61

1.36

1.70

1.34

Macroelements (g/kg)

K

5.4

1.7

7.0

2.2

Ca

8.1

9.8

7.6

9.3

Mg

2.1

1.7

2.1

1.5

Microelements (mg/kg)

B

25

10.8

29

13.2

Cd

< 2

< 2

< 2

< 2

Cu

82.2

116

23.6

35

Na

310.4

325.6

311.8

310

Pb

< 6

< 6

< 6

< 6

Hg

< 0.040

0.044

0.064

0.114

Macro and microelements were analyzed in duplicate; the standard deviation was < 10%. Data are given as mean ± standard deviations.

-Why was hop in pellet form turned into granules by spray drying?

The pellets of hop were ground to a powder (please see section 2.1 Raw material) to perform the extraction with ultrasound assisted extraction and pressurized hot water extraction (in sequence). The liquid extract was characterized and the fraction with potential interest due to their antioxidant activity was used to formulate microparticles. Spray drying is a technique suitable to produce microparticles (spheric particles) from aqueous solution, several articles found polymeric microparticles a preservative function for the valuable compounds (10.1016/j.focha.2023.100364).  

-Why was the effect of hops, which is widely used in brewing, on the humulene and lupulene contents not investigated? Maybe the granule structure of spray drying may be good, but once the components that give flavor to the beer disappear, the properties of the granule structure are meaningless.

Your comment is very interesting. However, in our case the objective of the work was to explore the phenolic fraction, which is less explored that that of other components of interest in the brewing industry. We have focused on the antioxidant activity found in these extracts, because it could be used in other food industries not only for brewing. The proposed characterization would have been of interest and will be performed in future studies.

The authors are grateful for your effort to help us to improve the manuscript. 

Reviewer 2 Report

The article titled "Microparticles by Spray Drying from Humulus lupulus antioxidants fractions within an Emerging Green Extraction Framework" discusses the extraction of bioactive compounds from two hop varieties and their encapsulation within microparticles. The article touches upon multiple important themes in current research, from green extraction to the microencapsulation of bioactives.  

Below are some technical points of improvement for the article:

The title could be made more concise and clearer.

Consider rephrasing to something like: "Microparticle Production by Spray Drying: Extracting Antioxidants from Humulus lupulus within a Green Extraction Framework." The title suggests the topic is about producing microparticles using spray drying from antioxidants derived from hops. Ensure that your objective, methods, and conclusions align with this. Right now, there’s a lot of focus on the extraction methods which, while relevant, seem to overshadow the microparticle production. The title could be revised for better clarity: "Green Extraction and Microencapsulation of Antioxidants from Humulus lupulus Varieties via Spray Drying". It might be helpful to clarify the significance of the phenolic content percentages.

How do these numbers compare to other extraction methods or other sources of antioxidants?

Consider mentioning the specific applications or benefits of the formulated microparticles in the conclusion of the abstract.

In line 30, instead of saying "control as drug delivery", perhaps use "modulate drug delivery".

In line 41, it's "Humulus lupulus", not "Humus lupulus". There's a mix of terminologies used: "green extraction", "eco-friendly extraction", and "alternative technology". More context could be provided about why this specific sequence (UAE followed by PHW) was chosen. This would provide readers with an understanding of the rationale behind the methodology.

It might be useful to define technical terms and abbreviations upon their first use, especially if they are not common knowledge among all readers. For instance, "UAE" (ultrasound-assisted extraction) and "PHW" (pressurized hot water extraction) should be defined early in the text. It might be helpful to stick to one term for consistency. Emphasize more clearly the unique properties or benefits of the hop-derived compounds compared to other sources.

In the "Raw material" section, provide details on how the hop samples were stored, i.e., temperature and humidity conditions.

For the "Ultrasound-assisted extraction" and "Pressurized hot water extraction" sections, specify the total duration of each extraction process. Clearly outline the advantages of using a sequential extraction process over single-step processes. Elaborate on why ultrasound-assisted extraction (UAE) and pressurized hot water extraction (PHW) were chosen. What advantages do they offer individually and in combination? The "Sequential extraction process" might be better structured if each extraction method is described under its subheading.

For "Characterization of microparticles", it's essential to provide a more detailed procedure on how the microparticles were formulated. Did you use any specific ratios? What was the drying time? Were there any solvents used?

In the "Rheological measurements" section, it would be helpful to specify why the 2% dispersion was selected for measurements. Ensure consistency in measurement units. For example, use either µm or nm throughout to represent particle sizes, rather than switching between the two.

The flow of information seems dense in some areas. Consider breaking long paragraphs into smaller ones for easier reading. Refrain from using repetitive terms and sentences. For instance, the mention of "sequential extraction" is repeated multiple times in adjacent lines. Ensure consistency in terminology. For instance, you’ve interchangeably used 'H. lupulus' and 'hop'. While technically correct, you should aim for uniformity in its usage.

Figure 1 and Figure 2: Ensure they are clearly labeled and easy to understand. The accompanying explanations should explicitly tie back to the objectives of the study.

Table 1: Consider including a summary or key findings immediately after the table to help readers quickly grasp the main insights.

The differences between the two hop varieties should be highlighted and discussed in depth. More emphasis on why antioxidant activity was chosen as the measure would be beneficial. Discuss the implications of the size differences in microparticles between the two hop varieties. Why might one be preferred over the other in certain applications? Dive deeper into the implications of your findings. For instance, why might the mannitol percentage be a limiting factor in spray drying yield?

Compare and contrast the results more extensively with other studies in the literature. Where do your findings align, and where do they diverge? While the conclusion is well-structured, consider delving into broader implications. What could the industry or academia take away from your findings? Clarify the "other fraction" mentioned in the conclusions. This appears abruptly without clear prior context.

Moderate editing of English language required

Author Response

Reviewer 2

The article titled "Microparticles by Spray Drying from Humulus lupulus antioxidants fractions within an Emerging Green Extraction Framework" discusses the extraction of bioactive compounds from two hop varieties and their encapsulation within microparticles. The article touches upon multiple important themes in current research, from green extraction to the microencapsulation of bioactives.

Below are some technical points of improvement for the article:

The title could be made more concise and clearer.

Consider rephrasing to something like: "Microparticle Production by Spray Drying: Extracting Antioxidants from Humulus lupulus within a Green Extraction Framework." The title suggests the topic is about producing microparticles using spray drying from antioxidants derived from hops. Ensure that your objective, methods, and conclusions align with this. Right now, there’s a lot of focus on the extraction methods which, while relevant, seem to overshadow the microparticle production. The title could be revised for better clarity: "Green Extraction and Microencapsulation of Antioxidants from Humulus lupulus Varieties via Spray Drying". It might be helpful to clarify the significance of the phenolic content percentages.

Thank you for your comment, following your advice the authors decided modify the title, the new title for the manuscript is “Green Extraction of Antioxidant fractions from Humulus lupulus Varieties and Microparticles Production via Spray Drying”.

How do these numbers compare to other extraction methods or other sources of antioxidants?

The comparison of the results was performed using references with similar origin of the raw materials, as vegetable for example, and articles about brewing industry. For instance, on lines 267-269: “Several works have found differences in bioactive and biological properties of the extracts using different extraction technologies [10,14,18].” where the references are:

[10] Olivares-Galván, S.; Marina, M.L.; García, M.C. Extraction of Valuable Compounds from Brewing Residues: Malt Rootlets, Spent Hops, and Spent Yeast. Trends Food Sci Technol 2022, 127, 181–197.

[14] Sanz, V.; Torres, M.D.; López Vilariño, J.M.; Domínguez, H. What Is New on the Hop Extraction? Trends Food Sci Technol 2019, 93, 12–22.

[18] Silva, G.V.A.; Arend, G.D.; Zielinski, A.A.F.; Di Luccio, M.; Ambrosi, A. Xanthohumol Properties and Strategies for Extraction from Hops and Brewery Residues: A Review. Food Chem 2023, 404.

Consider mentioning the specific applications or benefits of the formulated microparticles in the conclusion of the abstract.

It was added.

In line 30, instead of saying "control as drug delivery", perhaps use "modulate drug delivery".

It was modified.

In line 41, it's "Humulus lupulus", not "Humus lupulus". There's a mix of terminologies used: "green extraction", "eco-friendly extraction", and "alternative technology". More context could be provided about why this specific sequence (UAE followed by PHW) was chosen. This would provide readers with an understanding of the rationale behind the methodology.

Following you comment, the terminologies used was unified to green extraction and line 66 was modified.

According to the suggestion about the specific sequence UAE and PHW, the authors revise the introduction section found these sentences “Ultrasound assisted extraction or also known as sonication is used as green extraction technology in order to obtain bioactive compounds from hops. This technology can promote the release of bioactive compounds, whereas the application of hydrothermal treatments leads to enhanced phenolic extraction yield [21]. Other authors evaluated the cascade combination of ultrasound and microwave assisted extraction to obtain phenolic compounds [22].”.

In this case, other work where ultrasound assisted extraction with microwave assisted extraction was explorer was the base to prepare the current work. In this previous work, ethanol was used as solvent while in the current work only water was used as extractive agent.

It might be useful to define technical terms and abbreviations upon their first use, especially if they are not common knowledge among all readers. For instance, "UAE" (ultrasound-assisted extraction) and "PHW" (pressurized hot water extraction) should be defined early in the text. It might be helpful to stick to one term for consistency. Emphasize more clearly the unique properties or benefits of the hop-derived compounds compared to other sources.

Thank you for your advice, UAE and PHW was defined in the abstract. Following your comment, it was added also in the introduction section. Please, see on lines 62-65: “Green extraction technologies like pressurized hot water (PHW), microwave or ultrasound assisted extraction (UAE), have several advantages including saving time, energy and resources allowing high efficiency and low environmental impacts [19, 20]”

In the "Raw material" section, provide details on how the hop samples were stored, i.e., temperature and humidity conditions.

The raw material section was modified adding the room temperature and humidity conditions.

For the "Ultrasound-assisted extraction" and "Pressurized hot water extraction" sections, specify the total duration of each extraction process. Clearly outline the advantages of using a sequential extraction process over single-step processes. Elaborate on why ultrasound-assisted extraction (UAE) and pressurized hot water extraction (PHW) were chosen. What advantages do they offer individually and in combination? The "Sequential extraction process" might be better structured if each extraction method is described under its subheading.

In the section 2.2, the subheading “Ultrasound assisted extraction” include this paragraph “The operational conditions were: frequency at 80 kHz, and amplitude at 100%, in sweep mode, the extraction time for Perle was 30 min and for Nugget was 120 min, and also the temperature of the extraction was 55 ± 2 °C.” where the time of extraction was defined and specified.

In the subheading “Pressurized hot water extraction” a sentence was included, please see on lines 104-105: “When the reactor achieves the selected temperature (190 °C), it was cooled quickly until room temperature to open it.”

In section 2.2, before the explanation of UAE and PHW a sentence was added, please see on lines 86-88: “The sequential extraction process allows recovery the maximum quantity of bioactive compounds obtaining different fraction with different possible applications and an exhausted solid.”

The selection of UAE and PHW was added previously in the introduction section, please see above.

The methods of extraction were described under its subheadings.

For "Characterization of microparticles", it's essential to provide a more detailed procedure on how the microparticles were formulated. Did you use any specific ratios? What was the drying time? Were there any solvents used?

In the current work, no specific ratios were used, only to the crude liquid extract was added mannitol in several percentages (1, 2, 5 and 10%, w/w) to evaluate the effect of the polymer in the formulation of the microparticles.

The drying time was included in the operation conditions, the flow is 0.7 mL/min, for example if you use 20 mL of sample, the time you need is around 15 minutes. The time of drying is dependent of the quantity of aqueous solution you introduce in the equipment.

The unique solvent used in all the work is water, no more solvents were tested.

In the "Rheological measurements" section, it would be helpful to specify why the 2% dispersion was selected for measurements. Ensure consistency in measurement units. For example, use either µm or nm throughout to represent particle sizes, rather than switching between the two.

It was carefully revised and clarified.

The flow of information seems dense in some areas. Consider breaking long paragraphs into smaller ones for easier reading. Refrain from using repetitive terms and sentences. For instance, the mention of "sequential extraction" is repeated multiple times in adjacent lines. Ensure consistency in terminology. For instance, you’ve interchangeably used 'H. lupulus' and 'hop'. While technically correct, you should aim for uniformity in its usage.

Thank you for your advice, sentences were revised to improve the reading of the manuscript. Also, the mentions to “sequential extraction” were revised.

Figure 1 and Figure 2: Ensure they are clearly labeled and easy to understand. The accompanying explanations should explicitly tie back to the objectives of the study.

According to your comment, this sentence was included on lines 250-252: “This work was focused on the sequential extraction using two varieties of hop as raw material, known as Perle and Nugget. The main aim of this strategy was to obtain the maximum quantity of bioactive compounds. An overview of the processing and the main characterization of both materials is presented in Figure 1.”

Also, for Figure 2 the authors modified the explanation text to improve the understanding of it.

Table 1: Consider including a summary or key findings immediately after the table to help readers quickly grasp the main insights.

Following your comment, it was modified to clarify and help readers to understand. Please see on lines 283-285.

The differences between the two hop varieties should be highlighted and discussed in depth. More emphasis on why antioxidant activity was chosen as the measure would be beneficial. Discuss the implications of the size differences in microparticles between the two hop varieties. Why might one be preferred over the other in certain applications? Dive deeper into the implications of your findings. For instance, why might the mannitol percentage be a limiting factor in spray drying yield?

Following you comment, the whole manuscript was revised to clarify these issues. For instance, see on lines 433-440 and 466-468.

In this work, only mannitol was the polymers used to explorer its potential as carrier. Analyzing the results, when percentage increase the production yield of the microparticles decrease, for this reason, in a future work evaluate the behavior of several natural polymers could be a possible aim. In the text, this issue was clarified, please see on lines 435-436.

Compare and contrast the results more extensively with other studies in the literature. Where do your findings align, and where do they diverge? While the conclusion is well-structured, consider delving into broader implications. What could the industry or academia take away from your findings? Clarify the "other fraction" mentioned in the conclusions. This appears abruptly without clear prior context.

Thank you for your comment, the whole manuscript was revised, and the conclusions section was modified to improve it.

The authors are grateful for your effort to help us to improve the manuscript. 

Round 2

Reviewer 1 Report

Corrections were made by author. All corrections were pointed out in text..Only, if the mineral values are given in separate table, better.

minor revision need

Reviewer 2 Report

Accept in present form

Minor editing of English language required